# Optimising Medicines Administration for Patients with Dysphagia in Hospital: Medical or Nursing Responsibility?

**DOI:** 10.3390/geriatrics5010009

**Published:** 2020-02-19

**Authors:** David J. Wright, David G. Smithard, Richard Griffith

**Affiliations:** 1Professor of Pharmacy Practice, School of Pharmacy, University of East Anglia, Norwich NR4 7TJ, UK; 2University of Greenwich, London SE9 2UG, UK; david.smithard@nhs.net; 3Consultant Physician, Queen Elizabeth Hospital, Woolwich, London SE18 4QH, UK; 4Senior Lecturer in Law, College of Human and Health Sciences, Swansea University, Swansea SA2 8PP, UK; richard.griffith@swansea.ac.uk

**Keywords:** dysphagia, medicines administration, formulation alteration

## Abstract

Dysphagia is common—not only associated with stroke, dementia, Parkinson’s but also in many non-neurological medical problems—and is increasingly prevalent in ageing patients, where malnutrition is common and pneumonia is frequently the main cause of death. To improve the care of people with dysphagia (PWD) and minimise risk of aspiration and choking, the textures of food and drinks are frequently modified. Whilst medicines are usually concurrently prescribed for PWD, their texture is frequently not considered and therefore any minimisation of risk with respect to food and drink may be being negated when such medicines are administered. Furthermore, evidence is starting to emerge that mixing thickeners with medicines can, in certain circumstances, significantly affect drug bioavailability and therefore amending the texture of a medicine may not be straightforward. Research across a number of hospital trusts demonstrated that PWD are three times more likely to experience medication administration errors than those without dysphagia located on the same ward. Errors more commonly seen in PWD were missed doses, wrong formulation and wrong preparation through medicines alteration. Researchers also found that the same patient with dysphagia would be given their medicines in entirely different ways depending on the person administering the medicine. The alteration of medicines prior to administration has potential for patient harm, particularly if the medicine has been designed to release medicines at a pre-defined rate or within a pre-defined location. Alteration of medicines can have significant legal implications and these are frequently overlooked. Dispersing, crushing or mixing medicines can be part of, or misconstrued as, covert administration, thus introducing a further raft of legislation. Guidance within the UK recommends that following identification of dysphagia, the ongoing need for the medicine should be considered, as should the most appropriate route and formulation, with medicines alteration used as a last resort. The patient should be at the centre of any decision making. Evidence suggests that in the UK this guidance is not being followed. This article considers the clinical and legal issues surrounding administration of medicines to PWD from a UK perspective and debates whether medicines optimisation should be the primary responsibility of the prescriber when initiating therapy on the ward or the nurse who administers the medicine.

## 1. Introduction

Swallowing problems can be difficult to manage, particularly in the acute medical setting. Staff are more aware of the issues around the provision of food and liquids. Consideration of medicine administration is often forgotten, yet the administration of medicines to PWD in the hospital setting is fraught with day to day difficulties. Medication errors are a major concern in PWD and are the responsibility of the prescriber, and dispenser and the person administering the medication. This paper discusses the clinical and legal issues in reference to practice in the United Kingdom.

## 2. Swallowing Problems

Swallowing problems (Oro pharyngeal Dysphagia) are common in older people [1,2], and are defined as difficulty in food or liquid passing from the mouth, through the pharynx to the oesophagus and onwards to the stomach. Studies have reported a prevalence of approximately 15% of community dwelling older adults [3,4,5,6]. Dysphagia is now recognized as a geriatric syndrome as it has a multitude of aetiologies and contributes to a general decline in functional ability [1,2]. The occurrence of dysphagia in older people where no underlying medical issue has been identified is often referred to as presbyphagia, and in the presence of frailty, sarcopenic dysphagia is frequently referred to [7]. The commonest medical problems contributing to dysphagia prevalence are neurological [8,9,10] (stroke, dementia, motor neuron disease and multiple sclerosis); other less recognized causes are rheumatological, cardiorespiratory and infections (Table 1). 

To swallow safely, several physiological processes need to occur in sequence. The bolus needs to be prepared and presented to the pharynx; whilst this is occurring, the larynx elevates and rotates forward to the base of the tongue (which has come backwards), the soft palate descends and the posterior wall of the pharynx moves forward. As the larynx elevates, the true and false vocal cords approximate and close the pharynx off and breathing ceases momentarily. The bolus passes through the pharynx and through an open upper oeosphageal sphincter. Once the bolus has passed, the system relaxes and returns to the resting state.

Dysphagia is important because of its association with poor functional status in many cases [4] and increased morbidity and mortality [11]. Malnutrition and dehydration are common in people with dysphagia due to poor access to food and fluids or poor intake secondary to modified diets and fluids [12]. It is considered likely that many older people admitted to hospital with pneumonia will have an underlying problem with dysphagia [13,14]. Despite this acknowledgement that 44%-55% of people admitted with pneumonia may well have dysphagia, and that dysphagia is a common sequelae of frailty decompensation or other comorbidities, its presence is not routinely screened for [15] and frequently poorly managed. 

Before a management plan can be formulated, whether the patient can swallow safely or not needs to be identified. In the clinical situation, this may be done informally by members of the multidisciplinary team observing a person eat or drink (Table 2). A more formal approach may be used such as the bed side swallow assessments e.g., (TOR-BSST or GUSS [16,17] routinely used in stroke care. TOR-BSST has been validated in older patients [18]. A rapid screening questionnaire [19,20] must have reasonable specificity and sensitivity to trigger a referral to the speech and language therapy (pathology) department. 

The management of dysphagia is complex and involves a multidisciplinary team (Table 2). Following assessment by a speech and language therapist (or other trained professional), a management plan will be devised which may include further instrumental assessment. Once the nature of the dysphagia has been ascertained, the most appropriate method and route of providing nutrition (food and liquid) will be advised [1,11]. If it has been established that the oral route is still appropriate then it is necessary to determine whether liquid and foods require texture modification to prevent aspiration and choking [21]. To standardise this process, an international standard has been developed [21]. 

Following acute stroke, many people will recover their ability to swallow safely spontaneously [12], and following frailty decompensation this may occur also. For others, a period of rehabilitation may be appropriate [22]. There is no uniform approach to swallowing rehabilitation and the approach will need to be tailored to the individual and their particular problem (Table 3).

## 3. Medicine Administration

Medication is an important component of medical care, particularly in older and in particular frail older people who may have been prescribed multiple medications for their complex health needs. It is not uncommon, in a hospital environment, for food and medication to be stopped if there is any hint of dysphagia whilst awaiting assessment by a speech and language therapist. When a person has significant dysphagia, modified diets (food and liquids) may be recommended [21] and prescribed by the speech and language therapist; many clinical staff give no thought as to the effect that this may have on the bioavailability and pharmacokinetics of orally taken medication. Where medications are to be continued, there is often no structured medication review undertaken with a pharmacist [23]. The lack of consideration regarding how medicines are to be given to people with dysphagia by the prescriber (usually a doctor) creates significant difficulties for those who are expected to administer them (usually nursing staff). UK guidelines for management of medicines in patients with dysphagia recommend that, at the point of diagnosis, the presence of dysphagia should be assessed for, and medicines should be reviewed for ongoing suitability for the oral route and for those which are still required the formulation should be considered [24].

The medication review will need to include and begin with the simple question “is this medication needed?” And if it is, does the formulation need to change, or can it be taken with thickened fluids, or thickened itself if in a liquid formulation?

Where a suitable licensed formulation is not available, unlicensed options such as tablet crushing or dispersing can be undertaken [24]. However, the risks associated with this practice require careful consideration at the point of prescribing. The guidelines are written to ensure that administering nurses and carers are not faced with having to make complex decisions regarding administration at the bedside as this should have already been carried out by the prescriber to ensure that the process was simplified, legal and safe.

There is a range of approaches possible (selecting an alternative formulation, crushing or dispersing the medicine or omitting the medicine altogether) and depending on the nurse’s experience and training the final decision will differ. All options have clinical risks and potentially negative consequences.

Whilst swapping formulation may seem to be the easiest option, administration via a different route or formulation can change the drug’s bioavailability and this has to be taken into account for medicines which are at either end of the dose range or those with small therapeutic windows. Crushing or dispersing tablets with no modification to their release profile can in itself create difficulties due to taste and stability. Furthermore, tablets and capsules are relatively inexpensive with respect to acquisition costs and switching to other formulations frequently results in an increase in cost. If the formulation is unlicensed, this can result in a very significant cost increase [25,26,27].

Switching to a liquid medicine may seem to be the most appropriate; however, its texture is an important factor and most are not provided with instructions on how to safely thicken or dilute them. Consequently, whilst they may be easier for the patient to swallow they may increase risk of aspiration and subsequent problems which can arise from this. With evidence appearing which suggests that thickeners can negate the effect of a medicine [25,26,27], the decision to thicken a liquid will often not be evidence based and will require careful monitoring of the patient to ensure that the treatment is still effective.

Obviously omitting doses is never ideal unless the patient requests it or it is necessary to protect them. Interestingly, the omissions seen in the observational research were sometimes unintentional [28]. With nurses choosing to disperse tablets or capsule contents which are not designed to be dispersed and can take a significant amount of time, the dispersion was frequently left to happen and then nurses, who became engrossed in other activities, failed to return. This left the medicines in a slurry form next to the patient’s bed and not administered.

The act of crushing or dispersing medicines can be a safe alternative providing the medicine is not formulated such that drug release is modified. Slow release medicines frequently contain more than one ‘usual’ dose of the drug and consequently releasing this all at once can result in overdose and has resulted in reformulation of modified release products such as Oxycodone to reduce overdose associated deaths [29]. Similarly, if a medicine has a coating to either protect it from the stomach’s acidic environment, protect the gastrointestinal tract from the medicine itself or to deliver the medicine further down the gastro-intestinal tract, then any disruption of this can potentially harm the patient.

## 4. Medication Errors

A large-scale observational study undertaken in one region of England across four different hospitals found that patients with dysphagia were three times more likely to experience medication administration errors than those without dysphagia [28]. Patients with dysphagia were more likely to experience errors in drug preparation, less likely to receive their medication at all and more likely to receive their medicine in the wrong formulation. Where tablets and capsules were modified prior to administration, and thereby rendered unlicensed, there was no evidence that this practice had been authorised by a prescriber and consequently the practice was, strictly speaking, in contravention of the Human Medicines Regulations [30]. Similar results were found when care home staff were observed administering medicines to residents with and without dysphagia [31].

Medication errors in people with dysphagia were believed to be largely due to the additional complexity of having to administer a medicine to a patient with dysphagia [28]. This was because the prescribed formulations were not designed for patients with dysphagia, for administration via an enteral tube or because the nurse did not have the information at hand to make an informed choice as to what the best option would be. These beliefs were reinforced when the same patient was observed receiving the same medicines by two different nurses and received them entirely differently [32]. One recommendation resulting from the large number of errors seen at the point of medicines administration to patients with dysphagia was that nurses should be regularly observed to identify and address any learning needs [33]. Consequently, the responsibility for the medication errors seen within patients with dysphagia was being placed with the administering nurse.

## 5. Legal Aspects of Medication Administration

The Human Medicines Regulations 2012, regulation 46 [30], generally requires that medicinal products for human use are supplied and administered in accordance with a marketing authorisation. This defines the medicine’s terms of use in a summary of product characteristics which outlines the indications, recommended doses, contraindications and route of the medicine. The marketing authorisation also reassures health professionals of the medicine’s efficacy, safety, and quality.

The Medicines and Healthcare Products Regulatory Agency (MHRA) recognises that there will be situations where there is a need to use a medicine outside the terms of use set out in the statement of product characteristics [34]. The MHRA defines this as ‘off-label’ use and accepts it might be necessary to the clinical need of a patient. ‘Off label’ use includes the use of the medicine to treat a condition not specified in the marketing authorisation, using the medicine with a patient group not covered by the marketing authorisation or administering the medicine via a route not specified in the statement of product characteristics. Altering the formulation of a medicine by crushing or dispersing is considered to be ‘assembly’ and consequently the final product is unlicensed.

Whilst off label use is not recommended by the MHRA, they acknowledge that it is not unlawful if responsibility is assumed by the prescriber and is to be preferred to the use of an unlicensed medicine that has not been assessed for quality, safety and efficacy [34].

There are increased risks to the patient as a result of using a medicine off-label or unlicensed and both the MHRA [34] and the General Medical Council [35] caution that a health professional who causes harm to a patient as a result of off-label or unlicensed use would be in breach of their duty of care and liable in negligence. To discharge that duty of care, health professionals must ensure that the decision to use a medicine off label or unlicensed is based on reliable evidence. The patient must be provided with sufficient information about the medicine to allow them to make an informed decision [36]. Consequently, a prescriber choosing to authorise the use of a medicine, either off-label or unlicensed, should seek advice before doing so, if they are at all unsure about the safety of this prescription. The obvious source of advice is either local medicines information or the ward pharmacist. Both will be used to information requests of this nature and will be able to provide advice regarding options available and safety in a timely manner. A non-prescribing nurse choosing to ‘unlicense’ a medicine without authorisation from a prescriber should also seek advice at this stage so that the prescriber request can be appropriately informed.

Interestingly, the act of dispersion or crushing could be seen as ‘covert administration’ as a third party such as a relative watching medicines be crushed or dispersed and then mixed in with food or even just water could interpret this as hiding the medicine prior to administration. ‘Covert administration of medicines’ occurs when the drug is given in a concealed form [37]. The result is that the person is unknowingly taking medication. If the medicine is being given to an individual without capacity, this could be construed as being concealed; therefore, the law and regulatory guidance regarding covert administration should be considered.

The Care Quality Commission (CQC) (2015), the statutory regulator of health and adult care services in England have raised concerns that ‘what should be a last resort’ [Covert administration] is often regarded as ‘normal practice’ carried out for the convenience of staff rather than the best interests of patients. The CQC warns that to covertly administer medicines without first exploring other options is unlawful and fined a care provider some £4000 for unsafe management of medicines including the unsafe use of covert administration [38].

The Nursing and Midwifery Council (NMC) also considers covert administration to be a measure of last resort. As a general principle, by disguising medication, the patient is being led to believe they are not receiving medication, when in fact they are. The NMC Code at standard 16.5 does not consider this to be good practice [39]. The Nursing and Midwifery Council removed a nurse from the professional register after she was found to have routinely hidden medicine in jelly babies, jam sandwiches and custard without consent or other authority [40].

The National Institute for Health and Care Excellence (NICE) [41] argue that, unless there is an immediate need, covert administration should only occur when a management plan is in place following a formal best interest meeting. The purpose of this meeting is to agree whether administering medicines covertly is a necessary last resort measure in the patient’s best interests and it should be attended by nursing staff, the prescriber and pharmacist and a person who can communicate the views and interests of the patient such as a family member, friend or independent mental capacity advocate.

The Court of Protection holds that the use of covert administration of medicines with adults who lack decision making capacity is a serious interference with a person’s right to liberty and a private life under articles 5 and 8 of the European Convention of Human Rights [42]. The use of medication covertly, whether for physical health or for mental health, always calls for close scrutiny that, for vulnerable, incapable patients in hospital, can be achieved by obtaining a deprivation of liberty safeguard standard authorisation under the Mental Capacity Act 2005, schedule A1 [43].

## 6. Legal Guidance from the Court of Protection

The Court issued the following guidance to assist in cases of covert administration of medicine and deprivation of liberty;
Where there is a covert medication policy in place to decide on the use of covert administration it must include full consultation between healthcare professionals and family.Administering medication covertly must be clearly identified within the care plan, assessment of deprivation of liberty and authorisation of a deprivation of liberty.If the standard authorisation is for longer than six months there should be clear provision for regular, monthly, reviews of the care plan.There should also be regular reviews involving the family, RPR and healthcare professionalsAny change of medication or treatment regime should trigger a review where the medication is covertly administered.Supervisory bodies and best interests’ assessors should consider placing appropriate conditions to the standard authorisation that ensure these guidelines are complied with.


If tablet crushing or dispersing is carried out in patients without capacity, then the covert administration law requires careful consideration to ensure that practitioners are not believed to be in contravention. The patient’s doctor would usually be the most appropriate healthcare professional to oversee and support this process.

## 7. Conclusions

It can be seen that dysphagia increases the likelihood of medication errors and if it is not considered at the point of prescription initiation by the prescriber on the older persons ward it can create significant problems for nurses administering their medicines. If nurses choose to crush or disperse tablets to make them easier to swallow this can be illegal, unsafe and misconstrued as covert administration in those without capacity.

Whilst dysphagia is not routinely screened for in older people’s wards and assessed in a timely manner along with concurrent medication review, the patient is going to be at increased risk, as is the nurse who administers the medicines. Nurses are healthcare professionals who are expected to make autonomous decisions; however, it would be better if they were not routinely placed in this situation when faced with administering medicines to patients with dysphagia.

Although the paper is written from a UK perspective, the clinical facts remain wherever the problem is identified and it is perhaps legislation which differs. UK legislation is based on that from the European Union and therefore many of the legal considerations translate across Europe. Most countries have licensing laws and again changing medicines or using them outside of the license will usually transfer responsibility to the individual who chooses that course of action. The main differences may be surrounding laws regarding covert administration and assessment of patient capacity; however, it would be deemed unethical in most countries to administer medicines to a patient without their express consent. Consequently, alternative approaches to obtaining consent are required when individuals are unable to make decisions for themselves.

Considering the complexity of managing medicines in PWD, we therefore believe that ultimately it is the responsibility of the prescriber to routinely identify dysphagia in older persons, request that its nature is assessed by a speech and language therapist and then prescribe in such a manner that it minimises risk for all concerned. In order to achieve this, the ward pharmacist or local medicines information service should be used to identify the options available and the final decision made in discussion with the patient to identify their preferences. Where the patient does not have capacity, then appropriate members of the family should be included within this to ensure that it is not misconstrued as covert administration.

If the nurse is faced with administering medicines to a patient with dysphagia whereby the medicines have not been reviewed and options discussed with the patient or relative, then they should contact the pharmacist or local medicines information. Armed with evidence and options available, they should then speak to the patient and prescriber to decide how best to prescribe and administer the medicines.

In summary, therefore, dysphagia is everyone’s problem and is best managed by the multi-disciplinary ward team with the patient at the centre of all decision making.

## Figures and Tables

**Table 1 geriatrics-05-00009-t001:** Aetiology of dysphagia.

Neurological	Non Neurological
Stroke	Cardiac Disease
Dementias	Respiratory disease
Multiple Sclerosis	Rheumatoid Arthritis
Motor Neuron Disease	Osteo Arthritis
Parkinson’s disease	Ankylosing Spondylosis
Head Injury	Scleroderma
Brain tumour	Sjogren’s Disease
	Intubation
	Frailty decompensation
	Malignancy
	Dry Mouth/ Xerostomia
	Poor mouth care
	Oral thrush
	Periodontal infection
	Psychological
	Loss of teeth
	Medication

**Table 2 geriatrics-05-00009-t002:** Members of the multidisciplinary Team.

Multidisciplinary Team
Patient
Family
Paid Carer
Speech and Language Therapist
Nursing staff
Dietitian
Chef
Physiotherapist
Pharmacist
Doctor

**Table 3 geriatrics-05-00009-t003:** Rehabilitation techniques and management options.

Rehabilitation	Management
Tongue strength exercises (e.g., IOPI)	Modified diet
IQORO^TM^	Thickened Fluids
Vital Stim	
AMP Care	Postural Manoeuvres
Pharyngeal Stimulation	
Chin Tuck Against Resistance	Parenteral Nutrition
Shaker manoeuvre	
Laryngeal Resistance	Enteral Nutrition
McNeil Programme	
EMG Feedback	
Transcranial Magnetic Stimulation	
Mirror Neurons

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
