# Peer review of "Optimising Medicines Administration for Patients with Dysphagia in Hospital: Medical or Nursing Responsibility?"

_geriatrics, 2020, doi:10.3390/geriatrics5010009_

Round 1

Reviewer 1 Report

Your manuscript was interesting, and I think this theme would be more important for patients with dysphagia in the future.

Author Response

We have made change sto the paper as suggested by the reviewer. We thank reviewer for their suggestions, but as this is not a paper about the aetiology of dysphagia we have not included them. We have included more references regarding the prevalence of dysphagia.

Reviewer 2 Report

The paper described a review and expert opinion on the decision making of medication administration for patients with dysphagia in hospitals. It would be of merit for the readers but could consider following improvement: 1. Abstract: Line 11-12 First sentence needs changing "in stroke... ageing patients"? Line 21-22 Check grammar of sentence "The difference in the types of error...". In general, the abstract can be shortened. 2. Line 95-96, could mention potential tools to be used to detect dysphagia, e.g. the Sydney swallowing questionnaire. 3. line 123-124, crushing tablets can be an issue for immediate release medicines too, for example due to bad taste, toxicity, contamination, stability 4. line 144-146, there is a confusion on off-label use and unlicensed medicines. Should give clear definition. 5. Line 155-156, grammar 6. I found the article is lack of novelty. Most information presented already been described in previous publications. The conclusion that the doctors should be the decision maker does provide some useful insight in giving clinical guidance. However, this conclusion is not sufficiently supported by the argument in the paper. Not enough of evidence/argument to support this. Lastly, although I think this conclusion can provide some clarity to clinical practice, it also isolate the roles in dealing with medication administration to dysphagia patients. It should be a multi-disciplinary approach with defined responsibility from all team members. The paper focused on doctors and nurses and really played down the roles of pharmacists and speech and language therapists.

Author Response

The paper described a review and expert opinion on the decision making of medication administration for patients with dysphagia in hospitals. It would be of merit for the readers but could consider following improvement:

Abstract: Line 11-12 First sentence needs changing "in stroke... ageing patients"?

Patients added

Line 21-22 Check grammar of sentence "The difference in the types of error...". In general, the abstract can be shortened.

Changed start of sentence to ‘Errors more commonly seen in PWD were..’

Have removed 10 words from the abstract to make it more succinct.

Line 95-96, could mention potential tools to be used to detect dysphagia, e.g. the Sydney swallowing questionnaire.

We have added a paragraph discussing various screening/assessment tools that are used in the clinical setting. The Sydney Swallowing Questionnaire and EAT-10etc are not routinely used in the clinical environment.

line 123-124, crushing tablets can be an issue for immediate release medicines too, for example due to bad taste, toxicity, contamination, stability

Statement added into line 152.

line 144-146, there is a confusion on off-label use and unlicensed medicines. Should give clear definition.

We have changed the text to make the distinction between unlicensed and off-label more clearly and where we originally state just ‘off-label’ we have also stated ‘unlicensed’.

Line 155-156, grammar

Sentence modified

I found the article is lack of novelty. Most information presented already been described in previous publications. The conclusion that the doctors should be the decision maker does provide some useful insight in giving clinical guidance. However, this conclusion is not sufficiently supported by the argument in the paper. Not enough of evidence/argument to support this.

We disagree with the author as no other papers, to our knowledge, consider responsibility for the process.

Lastly, although I think this conclusion can provide some clarity to clinical practice, it also isolate the roles in dealing with medication administration to dysphagia patients. It should be a multi-disciplinary approach with defined responsibility from all team members.

The paper focused on doctors and nurses and really played down the roles of pharmacists and speech and language therapists. 

This is the final two paragraphs in the paper where all professions and the patient are clearly mentioned and are stated to be of importance in the process.

We therefore believe that ultimately it is the responsibility of the prescriber to routinely identify dysphagia in older persons, request that its nature is assessed by a speech and language therapist and then prescribe in such a manner that it minimises risk for all concerned.  In order to achieve this the ward pharmacist or local medicines information service should be used to identify the options available and the final decision made in discussion with the patient to identify their preferences.  Where the patient does not have capacity, then appropriate members of the family should be included within this to ensure that it is not misconstrued as covert administration. 

If the nurse is faced with administering medicines to a patient with dysphagia whereby the medicines have not been reviewed and options discussed with the patient or relative, then they should contact the pharmacist or local medicines information. Armed with evidence and options available they should then speak to the patient and prescriber to decide how best to prescribe and administer the medicines.

We have however added the following sentence:

In summary therefore dysphagia is everyone’s problem and is best managed by the multi-disciplinary ward team with the patient at the centre of all decision making. 

Reviewer 3 Report

The authors provides an interesting dissertation about the responsibilities for administrating medicines to dysphagic patients in hospitals.
However, I would suggest to fix the following major/minor concerns:

-)The authors should provide a more detailed introduction about dysphagia, by describing more estensively the causes (e.g., stroke is only one of the causes, apraxia is another one), and the general clinical/rehabilitation approach that is pursued in parallel to medicine administration. Only after providing the overall scenario, I would introduce the topic of the paper.
I suggest to include the following references:

Tibbling, L.; Gustafsson, B. Dysphagia and its consequences in the elderly. Dysphagia 1991. Finestone, H.M.; Greene-Finestone, L.S. Rehabilitation medicine: 2. Diagnosis of dysphagia and its nutritional management for stroke patients. CMAJ 2003. Yun, Y.J.; Na, Y.J.; Han, S.H. Swallowing apraxia in a patient with recurrent ischemic strokes: A case report. Medicine 2019, 98, e17056. McKenna, V.S.; Zhang, B.; Haines, M.B.; Kelchner, L.N. A systematic review of isometric lingual strength-training programs in adults with and without dysphagia. American Journal of Speech-Language Pathology 2017. Milazzo, M.; Panepinto, A.; Sabatini, A.M.; Danti, S. Tongue Rehabilitation Device for Dysphagic Patients. Sensors 2019, 19, 4657. Rogus-Pulia, N.; Robbins, J. Approaches to the rehabilitation of dysphagia in acute poststroke patients. Seminars in Speech and Language 2013.

-) I do not understand the label "main text" and the choice of a specific paragraph for the Court of Protection Guidance. I would suggest a more traditional layout (e.g., Introduction - Methods - Discussion - Conclusions)

-)The authors should provide a paragraph to explain the approach they pursued. Specifically, since they reviewed the scenario in England, I would suggest to make it clear, stating the methodology even in the abstract

.
-)The results must be organized by topics-subsections: possible errors in administrating medicines and consequences on patients (I would use also statistical references if available), Legislation references and guidelines. Please add tables and schemes for summarizing and discussing the results.

-) The english case is interesting but I would be interested in understanding how the results discussed in the manuscript can be applied/compared to the global scenario, especially in terms of legislation.

Minor concerns:
-)References are in the wrong format
-) I would suggest an overall check of the grammar and style.

Author Response

1)The authors should provide a more detailed introduction about dysphagia, by describing more estensively the causes (e.g., stroke is only one of the causes, apraxia is another one), and the general clinical/rehabilitation approach that is pursued in parallel to medicine administration. Only after providing the overall scenario, I would introduce the topic of the paper.
I suggest to include the following references:

Tibbling, L.; Gustafsson, B. Dysphagia and its consequences in the elderly. Dysphagia 1991. Finestone, H.M.; Greene-Finestone, L.S. Rehabilitation medicine: 2. Diagnosis of dysphagia and its nutritional management for stroke patients. CMAJ 2003.

Yun, Y.J.; Na, Y.J.; Han, S.H. Swallowing apraxia in a patient with recurrent ischemic strokes: A case report. Medicine 2019, 98, e17056.

McKenna, V.S.; Zhang, B.; Haines, M.B.; Kelchner, L.N. A systematic review of isometric lingual strength-training programs in adults with and without dysphagia. American Journal of Speech-Language Pathology 2017.

Milazzo, M.; Panepinto, A.; Sabatini, A.M.; Danti, S. Tongue Rehabilitation Device for Dysphagic Patients. Sensors 2019, 19, 4657.

Rogus-Pulia, N.; Robbins, J. Approaches to the rehabilitation of dysphagia in acute poststroke patients. Seminars in Speech and Language 2013.

We thank the Reviewer for his suggestion. We have listed in a tabular format various causes of oropharyngeal dysphagia.  We have added a comment on the role of rehabilitation of the swallow. We have added a table detailing different rehabilitation techniques. Although it is important to mention swallow rehabilitation in the context of this paper, it is not the main subject of discussion.

2) I do not understand the label "main text" and the choice of a specific paragraph for the Court of Protection Guidance. I would suggest a more traditional layout (e.g., Introduction - Methods - Discussion - Conclusions)

This is a debate paper and not a research paper and consequently we have adhered to the requirements associated with this style of paper. 

3)The authors should provide a paragraph to explain the approach they pursued. Specifically, since they reviewed the scenario in England, I would suggest to make it clear, stating the methodology even in the abstract

We have now made this clear in the abstract

4)The results must be organized by topics-subsections: possible errors in administrating medicines and consequences on patients (I would use also statistical references if available), Legislation references and guidelines. Please add tables and schemes for summarizing and discussing the results.

See response to point 2 above

5) The English case is interesting but I would be interested in understanding how the results discussed in the manuscript can be applied/compared to the global scenario, especially in terms of legislation.

The following has been added in response to this:

Although the paper is written from a UK perspective, the clinical facts remain wherever the problem is identified and it is perhaps legislation which differs. UK legislation is based on that from the European Union and therefore many of the legal considerations translate across Europe.  Most countries have licensing laws and again changing medicines or using them outside of the license will usually transfer responsibility to the individual who chooses that course of action.  The main differences may be surrounding laws regarding covert administration and assessment of patient capacity however it would be deemed unethical in most countries to administer medicines to a patient without their express consent.  Consequently, alternative approaches to obtaining consent are required when individuals are unable to make decisions for themselves.

Minor concerns:
6)References are in the wrong format

This has been corrected

Changed to ACS style

7) I would suggest an overall check of the grammar and style.

This has been done

Round 2

Reviewer 2 Report

Thanks to the author for the response and revision. I am satisfied with most of the changes and responses. There are a few minor considerations: Abstract, first sentence, suggest change to "... not only associated with stroke, dementia... and is increasingly prevalent in ageing patients. line 43, delete "people with" line 47, what is "Great Britian and Northern Island and England and Wales"? line 49, Oro-pharyngeal Dysphagia line 78: "may be used" line 79, only has half of the blanket mark. line 89, suggest change to "Whether the oral route is still appropriate.. line 91, is the "2 " at the end for reference? line 92, it is odd to use the question to start the paragraph line 93, reference citation style line 104, check sentence grammar. It is also very strong to say "no thought is given as to the effect that this may have on the bioavailability adn ... or orally taken medication. There are studies to look into this effect. line 113, only half of the quotation mark In general, the manuscript needs a thorough check in English and formating, especially the newly added information.

Author Response

All changes made as per reviewer request

Reviewer 3 Report

The authors properly addressed all the major/minor concerns. I would suggest a final review of the manuscript to fix minor typos/grammar errors and re-think of the layout of the tables in order make them more readable.

Author Response

This has been done
